# Luciferase-induced photoreductive uncaging of small-molecule effectors

Eric Lindberg[1], Simona Angerani[1], Marcello Anzola[1] & Nicolas Winssinger [1]

Bioluminescence resonance energy transfer (BRET) is extensively used to study dynamic systems and has been utilized in sensors for studying protein proximity, metabolites, and drug concentrations. Herein, we demonstrate that BRET can activate a ruthenium-based photocatalyst which performs bioorthogonal reactions. BRET from luciferase to the ruthenium photocatalyst is used to uncage effector molecules with up to 64 turnovers of the catalyst, achieving concentrations >0.6 µM effector with 10 nM luciferase construct. Using a BRET sensor, we further demonstrate that the catalysis can be modulated in response to an analyte, analogous to allosterically controlled enzymes. The BRET-induced reaction is used to uncage small-molecule drugs (ibrutinib and duocarmycin) at biologically effective concentrations in cellulo.

[1] Department of Organic Chemistry, NCCR Chemical Biology, Faculty of Science, University of Geneva, 30 quai Ernest Ansermet, 1211 Geneva, Switzerland. Correspondence and requests for materials should be addressed to N.W. (email: Nicolas.Winssinger@unige.ch)

Bioluminescence resonance energy transfer (BRET) has become an important tool for investigating dynamic interactions in living systems. BRET is based on energy transfer from a luciferase donor to a proximal fluorophore acceptor. This technology has found many applications for real-time monitoring of dynamic processes such as protein–protein and protein–ligand interactions and for other uses, including imaging ion concentrations in cellular assays and whole-animal imaging[1–7]. However, to date, it has essentially been limited to imaging applications. Recently, BRET was used to switch a photochromic fluorescent protein (Dronpa) to an ON state[8]. Concurrently, the first application extending BRET from imaging to a signaling event was reported[9–11]. In this case, BRET from *Gaussia* luciferase to rhodopsin ion channels (luminopsins) was used to modulate neuronal activity. Herein, we report a different approach wherein luciferase resonance energy transfer activates a ruthenium-based photocatalyst to perform chemoselective and bioorthogonal chemical transformations. Ru(bpy)$_3$Cl$_2$ [Tris(2,2'-bipyridyl)dichlororuthenium(II)] has previously been shown to photocatalytically and bioorthogonally reduce azides[12]. In previous studies, we have harnessed this chemistry using the closely related Ru(bpy)$_2$phen for nucleic acid–templated reactions in live cells and live vertebrates[13–16]. More recently, we showed that a pyridinium linker was reductively cleaved using the same ruthenium-based photocatalysis with the fastest rate for nucleic acid–templated reactions ($k_{cat}/K_M$ of $10^5$ M$^{-1}$s$^{-1}$)[17]. One of the brightest known luciferases, NanoLuc (NLuc), developed from *Oplophorus* luciferase by Promega, has an emission profile that extensively overlaps with the absorption spectrum of Ru(bpy)$_2$-phen (Fig. 1a)[18]. While the spectral overlap between the emission of NLuc and absorbance of Ru(bpy)$_2$phen suggests that resonance energy transfer between luciferase and ruthenium should proceed

**Fig. 1** Concept and design of LUPIN. **a** The emission spectrum of NLuc (blue) overlaps very well with the absorption spectrum of the ruthenium photocatalyst (red), suggesting that efficient BRET should be possible. **b** The fusion protein SNAP-Pro30-NanoLuc (NLuc)-cpDHFR is linked via self-labeling SNAP to the synthetic linker containing the ruthenium photocatalyst and methotrexate (purple ball, DHFR ligand), positioning the ruthenium in close proximity to the NLuc. Free methotrexate (green ball) can push the sensor into the open conformation, thus turning off the BRET to the photocatalyst. **c** By installing a PNA next to the ruthenium catalyst, complementary pyridinium substrates can bind and undergo photoreductive cleavage by the ruthenium, unmasking the effector molecule

when placed in the required proximity, a fast chemical transformation is required to capture this excited state and effectively translate the energy transfer into a reaction (Fig. 1b). To be useful in a cellular or in vivo setting, the photocatalysis should proceed with substrates at low concentration (µM). To this end, we introduce LUPIN (**lu**ciferase-based **p**hotocatalysis **i**nduced via **n**ucleic acid template), wherein the photocatalyst is conjugated to a nucleic acid (PNA) to create a high effective concentration of substrate capitalizing on the fast chemical transformation of nucleic acid-templated reactions at low substrate concentrations (Fig. 1c).

## Results

**Energy transfer from NLuc to ruthenium photocatalyst**. We envisioned a semi-synthetic system that would allow us to bring the ruthenium complex, NLuc, and the pyridinium substrate into close proximity of each other. The Johnsson lab has recently reported semi-synthetic sensors for monitoring drug concentrations (LUCID: luciferase-based indicator of drugs)[19,20]. LUCID is a dynamic platform with three components fused together: a SNAP protein to conjugate a synthetic linker containing the dye–drug adduct, NLuc for bioluminescence, and a receptor protein that binds the drug. This construct responds to a drug by changing the proximity of the fluorophore to NLuc, and a LUCID developed for methotrexate (MTX) was shown to respond with an $EC_{50}$ of 85 µM. Inspired by this precedent, we reasoned that an analogous linker containing the ruthenium complex should achieve efficient energy transfer from the NLuc to the ruthenium in an MTX-dependent manner. We further added a 5-mer PNA adjacent to the ruthenium complex for templated reactions (see Fig. 1 and Supplementary Fig. 1 for the explicit structure of the linker). After ligating our linker (BG-(Ru)(PNA)-MTX: (**1**) to the protein construct (Supplementary Fig. 2), we measured the luminescence emission spectrum in the presence and absence of MTX, testing the efficacy of BRET (Fig. 2 and supplementary Figs. 3 and 4). We observed a new emission band around 610 nm, corresponding to the expected emission wavelength of the Ru (bpy)$_2$phen complex in the absence of MTX. Addition of MTX (100 µM) dramatically reduced the energy transfer from the NLuc to the ruthenium complex, concurring the results reported with LUCID[19] and consistent with the reported $EC_{50}$[19]. Based on the quantum yield of Nluc and the spectral overlap of the Nluc emission spectrum and the excitation spectrum of the Ru-complex, we calculated the Forster distance to be 16 Å. Since the energy transfer efficiency between NLuc and the Ru-complex was found to be 0.64, the average distance was estimated to be less than 16 Å.

**Photoreductive release of rhodamine with LUPIN**. Having established that the alignment of the ruthenium catalyst and NLuc allowed for energy transfer, we next investigated if this energy transfer could be translated into the reductive unmasking of a rhodamine with a pyridinium construct containing a 5-mer PNA, complementary to the PNA situated next to the photocatalyst (Fig. 1c). Thus 100 µM furimazine was added to the LUPIN sensor construct (50 nM) in the presence of 0.5 µM PNA-PyRho- substrate (**2**, Fig. 2b) with 10 mM sodium ascorbate as a stoichiometric reducing agent for ruthenium catalyst turnover. To our delight, we observed an increase in fluorescence due to the release of rhodamine, reaching 65 nM (1.3 turnovers, Fig. 2b), a slight excess relative to the LUPIN sensor. On the other hand, when supplementing the reaction with 100 µM MTX the yield was reduced by roughly half (35 nM rhodamine released), consistent with the shift in equilibrium of the sensor to an open conformation ($EC_{50}$ 85 µM), thus reducing BRET efficiency[19]. As

further controls, the reaction was performed with the SNAP-NLuc-DHFR construct pretreated with benzyl guanine to saturate SNAP prior to the addition of the BG-linker containing the photocatalyst-PNA-MTX. In this case, the synthetic linker will not be covalently attached to the sensor. The experiment was performed under the same conditions as above (without MTX). A marginal reaction was observed (which can be expected due to the photon flux from NLuc and absorption by the ruthenium photocatalyst; however, this should always be small compared to resonance energy transfer). Finally, we performed the reaction using the LUPIN sensor with a substrate lacking the PNA. We detected a dramatically reduced reaction. Taken together, these data indicate that the observed reaction is a product of resonance energy transfer from NLuc to the ruthenium photocatalyst and that the nucleic acid-templated reaction is important to increase the effective substrate concentration to achieve the desired chemical transformation. Furthermore, the rate of reaction can be allosterically modulated by shifting the conformation from the closed form to the open form with the addition of MTX (Supplementary Fig. 5). This first set of reactions effectively stopped after 15 min, which was attributed to the fast consumption of furimazine by NLuc. Thus, we decreased the concentration of the LUPIN sensor (50, 10, 2 nM) while keeping the furimazine substrate (100 µM) and PNA-PyRho (**2**) concentrations constant (5 µM; prior experiment performed at 0.5 µM). We compared the progression of rhodamine concentration to the luminescence decay of the unlabeled LUPIN sensor (Fig. 2c). At 50 nM LUPIN sensor, the luminescence decay was fast due to rapid furimazine consumption ($t_{1/2} \sim 8$ min) and the photoreduction reaction dramatically slowed down after 10 min, reaching a rhodamine concentration of 360 nM (TON of 7, Fig. 2c). At 10 nM LUPIN sensor, the observed luminescence half-life increased to ~21 min and the photoreduction reaction progressed to deliver 640 nM rhodamine (TON of 64 in 2 h, Fig. 2c). At 2 nM LUPIN sensor, the observed luminescence half-life increased even further to ~78 min with an observed unmasking of 130 nM rhodamine (TON 64 in 2 h, Fig. 2c). This increase in TON with a decrease in LUPIN sensor from 50 to 10 nM suggests that at high LUPIN sensor concentration (50 nM), the rate of bioluminescence far exceeds the rate of substrate photoreduction. It is noteworthy that the concentration of product obtained in the reactions (100–600 nM) is well within the range of the effective concentration of many drugs and that this can be achieved with a low concentration of LUPIN (2–10 nM). Further evaluation of our LUPIN system revealed there is a direct correlation between the photon flux emitted from Nluc and the amount of uncaged product (Supplementary Fig. 6a–d, reactions at different furimazine concentrations), as could be expected based on the reaction mechanism. The reaction was shown to be effective over a broad range of substrate concentrations (5–0.2 µM of substrate Supplementary Fig. 7a–c). It should be noted that 100 µM furimazine is not toxic[21]. We observed that the reaction rate did not change with decreasing NaAsc concentrations (Supplementary Fig. 8a–c), down to 0.1 mM ascorbate. The reduction of the photoexcited state of the ruthenium catalyst proceeds at a rate near diffusion control and this step is not limiting in the overall reaction progression. However, it should be noted that a high concentration of NaAsc (10 mM) is within the clinically established tolerance[22].

**Photocatalyzed uncaging of effector PNA prodrugs**. Having established that we could effectively release a fluorophore by ruthenium-catalyzed nucleic acid-templated pyridinium photoreduction, we next investigated whether LUPIN could also be used to release effector molecules at biologically effective concentrations. As examples we chose three different drug molecules

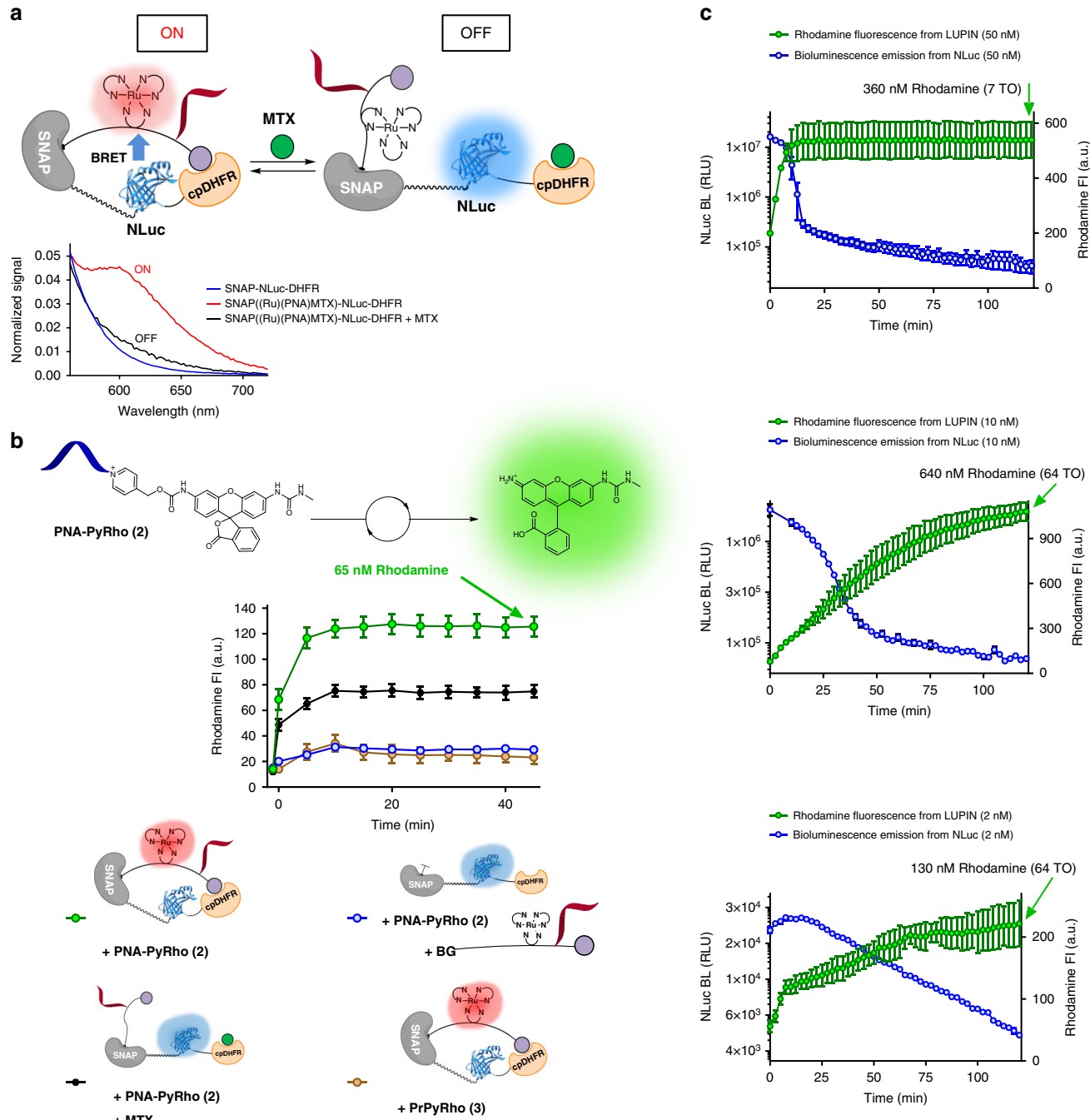

**Fig. 2** Characterization of LUPIN. **a** Bioluminescence resonance energy transfer from Nluc to ruthenium is observed when SNAP-NLuc-DHFR was labeled with BG-(Ru)(PNA)-MTX (**1**). Emergence of a new band at 610 nm indicates BRET from Nluc to ruthenium (red line); in the presence of methotrexate, more of the sensor shifts into the open conformation, decreasing BRET efficiency (black line). **b** Fluorescence enhancement due to rhodamine uncaging by the LUPIN system is observed. In the presence of methotrexate, the reaction is partially inhibited due to lower BRET efficiency. In the absence of nucleic acid template, or in the absence of a covalent link between the synthetic linker and the protein construct, the reaction is marginal. Reaction conditions: SNAP-NLuc-DHFR labeled with BG-(Ru)(PNA)-MTX (**1**) (50 nM), PNA-PyRho (**2**) or PrPyRho (**3**) (0.5 µM), sodium ascorbate (10 mM), furimazine (100 µM), and methotrexate (100 µM). BG benzyl guanine. **c** Varying the concentration of the labeled SNAP-NLuc-DHFR protein (50, 10, and 2 nM) has a significant impact on luminescence decay and the rate of rhodamine release

(Fig. 3): raloxifen, a partial agonist of the estrogen receptor;[23,24] ibrutinib, a potent covalent kinase inhibitor (IC$_{50}$ of 3–6 nM in chronic active BCR signaling B-cell lymphoma[25]); and a duocarmycin analog, a highly cytotoxic prodrug of a DNA-alkylating agent with IC$_{50}$ < 1 nM in human bronchial carcinoma cells[26]. We prepared drug conjugates by attaching each drug to the pyridinium linker and 5-mer PNA (Fig. 3, **6**–**8**). In all three cases,

a heteroatom involved in the binding to the target was selected for conjugation to the pyridinium linker, thus abrogating its designed activity and yielding a caged prodrug. The photocatalyzed uncaging of the prodrug was validated through a templated reaction using LED irradiation (Fig. 3). These experiments showed a clean conversion of the caged prodrug to the active drug within minutes of irradiation.

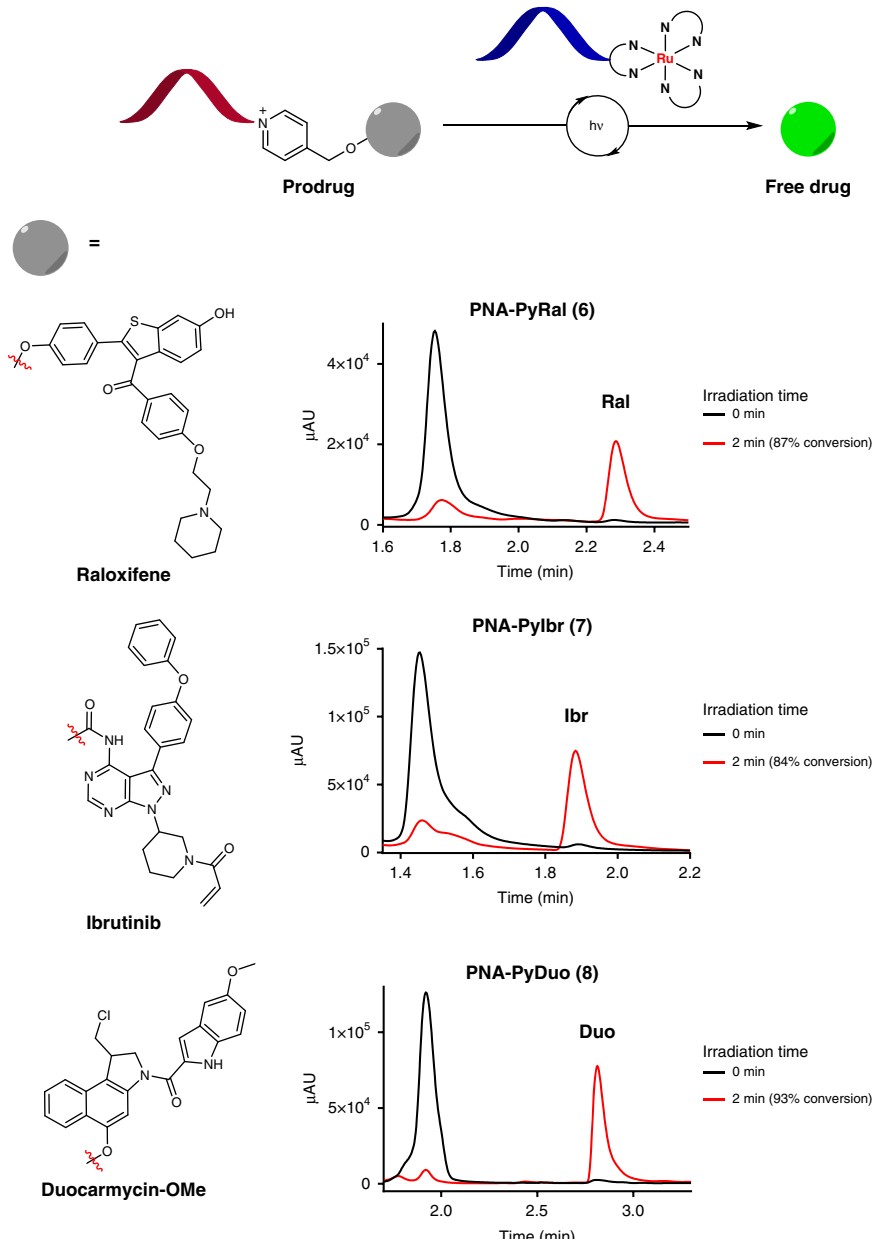

**Fig. 3** PNA-templated photoreductive release of effector molecules by ruthenium photocatalysis. Ru-PNA-5mer (**5**) (10 µM), PNA-Py-Drug (**6**–**8**) (100 µM), sodium ascorbate (10 mM) in PBS 1X(pH 7.4). Irradiation with 455 nm light before and after 2 min was followed by LC-MS quantification

**LUPIN release of ibrutinib**. We next investigated the LUPIN-promoted uncaging of ibrutinib, a covalent inhibitor of ErbB2. To assess the level of target engagement following LUPIN-induced uncaging, we also prepared a conjugate of ibrutinib with Cy3 (**9**, Fig. 4). We first validated that this conjugate labeled ErbB2 in SKBR3 cells, resulting in bright cellular fluorescence, and that saturation of the ErbB2 with free ibrutinib (1 µM, 30 min) resulted in a dramatic reduction of fluorescence (Supplementary Figs. 9 and 10). This observation is consistent with the fact that ibrutinib covalently engages its target and ligand exchange is not possible once the target is saturated with the drug. Next, we compared the level of labeling obtained in cells treated with the caged ibrutinib (PNA-Py-Ibr **7**, 10 µM) with and without LUPIN. In the absence of LUPIN, strong cellular labeling was observed, indicating that the caged drug could not engage the target, allowing the labeling to proceed. In the presence of LUPIN (10 nM), a dramatic reduction of labeling was observed,

indicating that sufficient drug had been uncaged to saturate the target in these SKBR3 cells (Fig. 4 and Supplementary Fig. 11). These results demonstrate that a drug can be unmasked through BRET-induced photocatalysis and that the technology should be broadly applicable to any small-molecule effector.

**LUPIN release of duocarmycin in MCF-7 cell culture**. Duocarmycin analogs have been used as prodrugs with diverse modes of uncaging[26–29]. As long as the phenol of a prodrug is engaged in a covalent bond, the cyclopropanation reaction leading to the formation of the active drug cannot take place. In the present case, the phenol is masked by a primary benzylic ether. In the course of evaluating the toxicity of the free drug versus the prodrug, we discovered that the linker suffered from low levels of background immolation that were sufficient to yield partial cytotoxicity given the extreme potency of the duocarmycin pharmacophore. Using

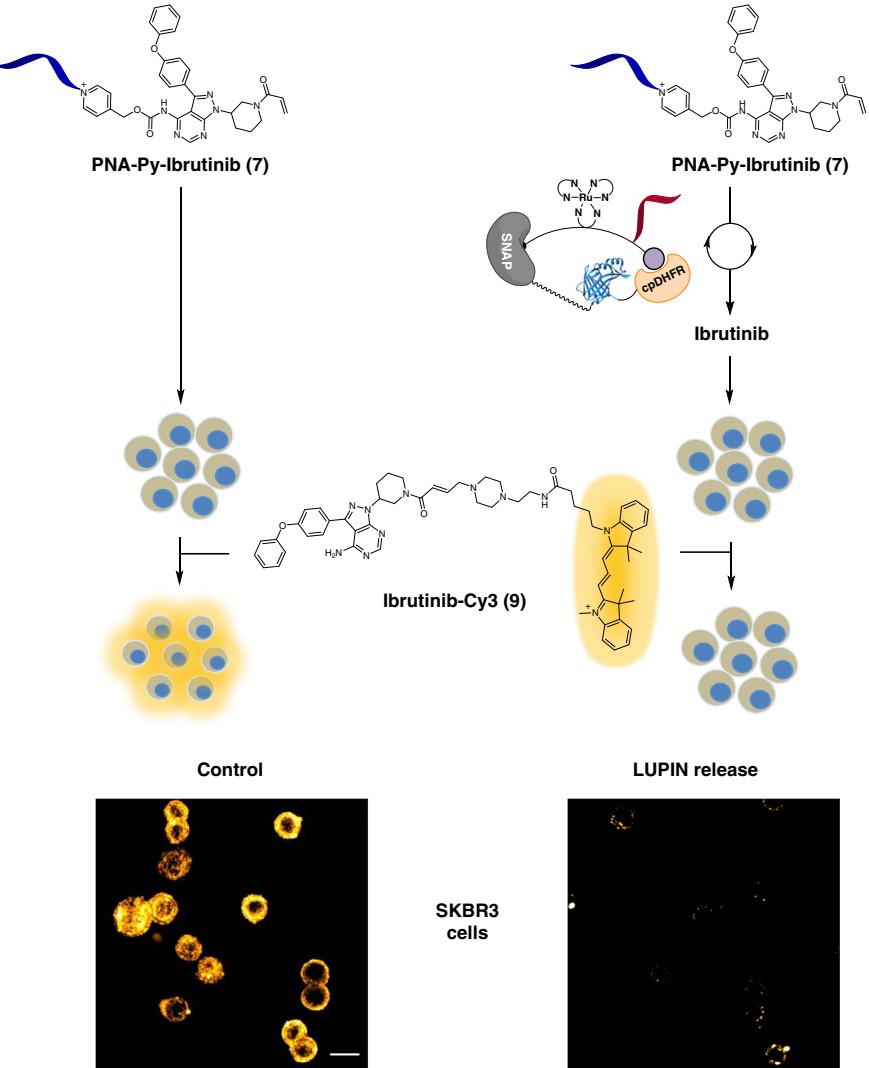

**Fig. 4** LUPIN release of ibrutinib. Ibrutinib release by LUPIN was demonstrated by competition with ibrutinib-Cy3 (**9**) derivative in SKBR3 cells overexpressing ErbB2. SKBR3 cells were treated with either the prodrug alone (**7**) or prodrug (**7**) + LUPIN for 30 min; then the cells were treated with ibrutinib-Cy3 (**9**). The lower fluorescence observed for the LUPIN-uncaged product population indicates that the ErbB2 was saturated with the uncaged prodrug. Conditions for LUPIN release: SNAP-NLuc-DHFR labeled with BG-(Ru)(PNA)-MTX (**1**) (10 nM), PNA-PyIbr (**7**) (5 μM), sodium ascorbate (10 mM), furimazine (100 μM), and Ibr-Cy3 (**9**) (50 nM). Scale bar: 20 μm

the caged rhodamine for a more quantitative analysis, we measured the half-life of the linker in media, and observed 0.6% immolation after 12 h ($k = 1.916 \times 10^{-7}\,s^{-1}$; Supplementary Fig. 12). Assuming a first-order rate of decomposition, the half-life of the linker is 1005 h. While this low level of immolation was not detected in previous experiments with shorter readouts, the extreme cytotoxicity of the uncaged duocamycin drug coupled to the longer assay period (3 days) reduced the therapeutic window. We pursued two strategies to reduce the background hydrolysis of the immolative linker: increasing steric bulk on the aryl ring using a 2,6-lutidine rather than a pyridine, and using a secondary benzylic ether in the connection to duocarmycin. Comparison of the templated reaction with the modified immolative linkers showed that both were competent in ruthenium-photocatalyzed immolations but the modification with the secondary benzylic position performed better than the lutidine (Supplementary Fig. 13). This modification led to comparable rates of reaction and immolation as the original primary benzylic pyridinium linker. However, no background hydrolysis was measured when using the latter linker (incubation of the substrate with media for 12 h did not afford a measurable reaction; Supplementary Fig. 13). We also

observed a two-fold improvement in reaction yield when using Leibovitz's medium instead of HEPES buffer (Supplementary Fig. 14), while the photon flux and luminescence decay profile of NLuc only changed marginally. With this new linker in hand, we compared the cytotoxicity of the caged prodrug to the uncaged drug. Using MCF-7 as a prototypical cancer cell line, strong discrimination of cytotoxicity was clearly apparent at low nM concentration, with the uncaged drug resulting in complete cellular death, while the caged drug resulted in comparable cellular density to a control without drug. A therapeutic window of >100-fold was measured at an optimal operating concentration of 5–25 nM (Supplementary Fig. 15). Next, we compared the cytotoxicity of the prodrug with LUPIN at various concentrations of furimazine. In the absence of furimazine, LUPIN did not trigger BRET and the prodrug remained caged. At all tested prodrug concentrations, there was a clear dose response of toxicity based on furimazine concentration (Fig. 5. and Supplementary Fig. 16a–d). In the absence of prodrug, no significant differences in cellular growth were observed in response to furimazine concentration up to 100 μM. With prodrug, the best results were obtained at a concentration of 6.25 nM. At this concentration, there was a dramatic

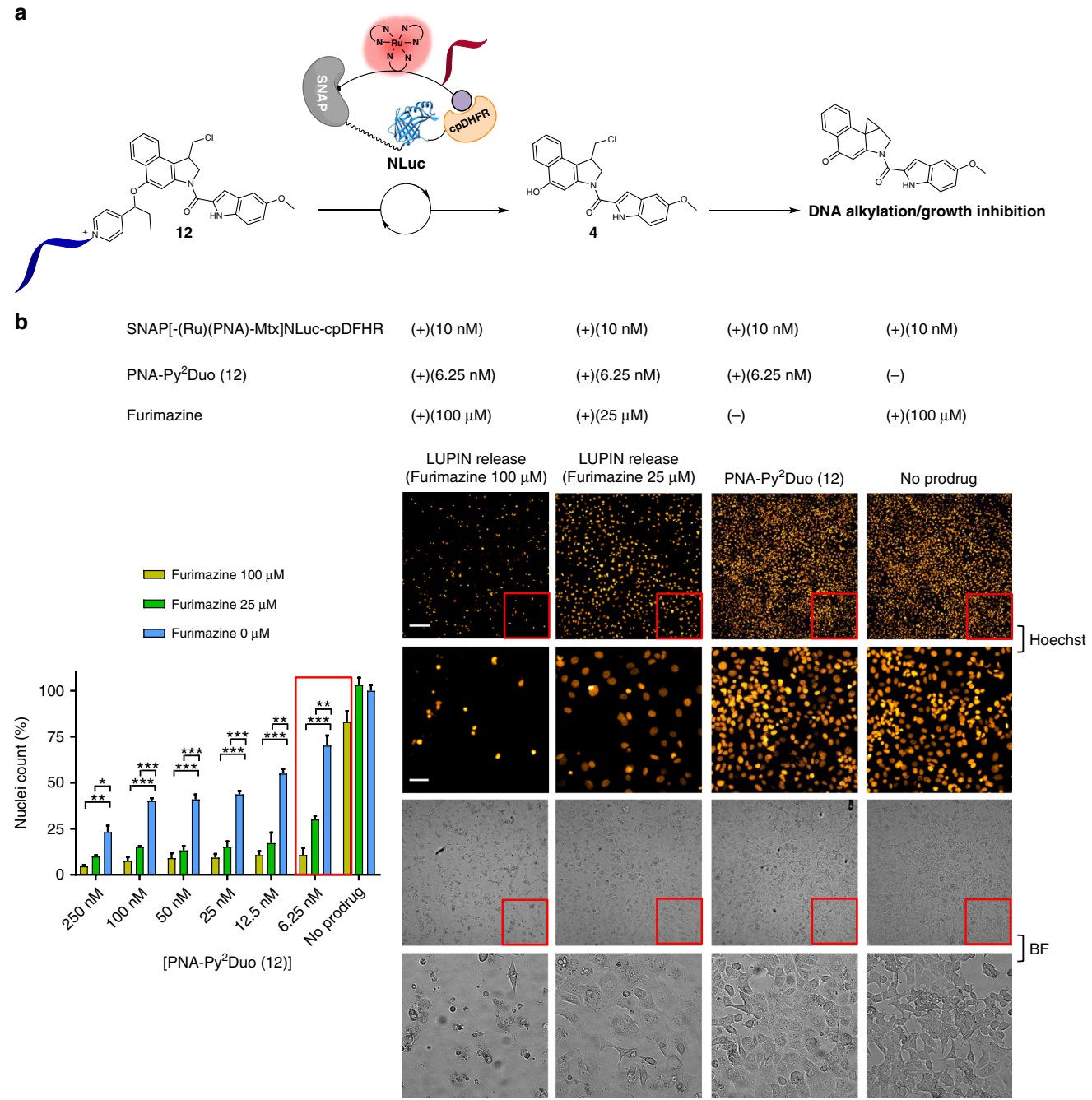

**Fig. 5** LUPIN release of cytotoxic duocarmycin analogue in MCF-7 cell culture. **a** Schematic representation of Duo-OMe (**4**) release by LUPIN with furimazine to generate DNA alkylating agent. **b** Quantification of cellular growth (MCF-7) by nuclei count. Bar graph of nuclei count across different furimazine and prodrug (PNA-Py$^2$Duo **12**) concentrations (left panel). Representative images of cells (right panel) across different prodrug (**12**) and furimazine concentrations imaged by fluorescence (Hoechst) and bright field (BF). The red square indicates the zoomed-in area below it. The bar graph is the average of three independent experiments ran in triplicates. Error bars show ±1 standard deviation from the mean. Statistics were calculated using a two-tailed *t*-test with unequal variances (Welch's unpaired *t*-test). *$p < 0.05$, **$p < 0.01$, ***$p < 0.001$. Scale bar: 200 μM; zoom-in scale bar: 50 μM

difference in experiments using 100 μM furimazine, which resulted in mostly dead cells in bright-field images with only a few nuclei left, versus experiments run under the same conditions but without furimazine, in which cells reached near confluence. Intermediate concentrations of furimazine (e.g., 25 μM) still led to growth inhibition and a partial cytotoxic response with more live cells remaining but also resulted in extensive debris from apoptotic cells (Fig. 5, inserts). The clear dose response observed as a function of furimazine concentration strongly suggests that uncaging of the prodrug proceeds through a BRET-photocatalysis

process. While the best results were obtained at 6.25 nM prodrug, this trend held over a concentration range of 6.25–250 nM prodrug (Supplementary Fig. 16a–d). The fact that the prodrug can still be uncaged at a concentration as low as 6.25 nM attests to the importance of the nucleic acid component in the design. This again demonstrates that our LUPIN system is capable of releasing effector molecules at biologically relevant concentrations.

The present design of LUPIN was conceived from an established semi-synthetic BRET sensor that was selected because it should be adaptable to BRET-induced photocatalysis and

offered a testing ground for allosteric responsiveness. The positive results obtained suggest that BRET-induced photocatalysis should be achievable with other bioluminescent proteins/photocatalyst pairs. There are exciting developments in bioluminescence imaging, with recent reports of brighter luciferases that could further improve the amplification and broaden the dynamic range of BRET-induced photocatalytic transformation. For instance, GLucM23 is a recently reported extremely bright *Gaussia* luciferase mutant[30], and teLuc[21], a NLuc mutant twice as bright as the native luciferase.

Conventionally, BRET has almost exclusively been used to monitor biochemical pathways and events. Herein, we have demonstrated that BRET can be harnessed to design a semi-synthetic biochemical pathway. It should be noted that the design is highly modular and lends itself to multiplexing. For instance, bioluminescent proteins using different substrate specificities could be functionalized with unique PNA tags to direct parallel chemistries without cross-talk. As our construct also responds to direct light irradiation, the system can be controlled both chemically and optically.

In summary, we report the first example of a BRET-induced bioorthogonal reaction. The reaction is promoted by an energy transfer from a bioluminescent protein (Nluc) to a ruthenium-based photocatalyst which uncages a substrate through a templated reaction. The reaction was demonstrated using a known BRET-based sensor with a novel synthetic linker. The sensor is composed of (DHFR) fused to a luciferase (NLuc) and a self-labeling protein (SNAP). The synthetic linker is composed of the ruthenium-based photocatalyst, a PNA strand for hybridization-based templated reactions, and the ligand that interacts with the receptor of the sensor. We demonstrated that templated chemistry was essential to achieve efficient uncaging of substrate. The nucleic acid template offers a flexible platform to tune the affinity of the semisynthetic construct for the substrate allowing reactions to proceed at low substrate concentrations (<1 μM). BRETs have frequently been utilized as reporting modalities in diverse sensors. The present work extends the utility of such sensor to respond with a reaction that uncages an effector molecule, as demonstrated with LUPIN. We showed that LUPIN operates at low concentration (2–50 nM) and is capable of catalytic substrate uncaging with up to 64 turnovers, yielding 100–600 nM product. This chemistry was used to unmask a selective kinase inhibitor (ibrutinib) and an extremely potent cytotoxic prodrug (duocarmycin analog). We expect this technology to be useful for logic-based responsiveness to environmental cues and prospective smart therapeutics. LUPIN overcomes the challenge of spatial resolution and light penetration in photoreactions. Furthermore, genetic encoding of the sensor can restrict the reaction to specific cell types. In a broader context, this chemistry opens new horizons in biotransformations, and we expect it to find applications in synthetic biology. While the present work was performed with a ruthenium-based photocatalyst, we anticipate that this concept can be extrapolated to any photocatalyst (transition metal-based or organic) with an absorption overlapping the emission of a bioluminescent protein.

## Methods

**Labeling of SNAP-Pro30-NLuc-cpDHFR with 1**. An aliquot of **1** (4 μM) was added to an Eppendorf tube containing a solution of SNAP-Pro30-NLuc-cpDHFR (1 μM) in HEPES (50 mM) NaCl (50 mM) (pH 7.2). The solution was left shaking at room temperature for 1 h. The labeling was monitored and confirmed via MALDI-TOF. Labeled proteins were purified using an ultra-centrifugal filter device with a 30 kDa MWCO (0.5 mL), performing three exchanges of 450 μL HEPES (50 mM) NaCl (50 mM) (pH 7.2) to remove excess amounts of **1**. Subsequent experiments, however, showed that non-covalently linked **1** had no effect. SNAP-Pro30-NLuc-cpDHFR labeled with **1** and purified via spin filtration showed no difference in performance in comparison with the product of a crude reaction

mixture. The experiments depicted in Fig. 2 were performed with material obtained via spin filtration while those illustrated in Figs. 4 and 5 were conducted with crude reaction mixture.

**Representative conditions for LUPIN release**. Furimazine (100 μM) was added to a solution containing SNAP-Pro30-NLuc-cpDHFR labeled with **1** (10 nM), PNA-PyRho (**2**) (5 μM), and NaAsc (10 mM) in HEPES (50 mM) NaCl (50 mM) (pH 7.2). The rhodamine fluorescence generated was quantified by comparison with free rhodamine standard curves to estimate the concentration of rhodamine released.

**Luminescence Spectra of SNAP-Pro30-NLuc-cpDHFR**. Luminescence spectra of constructs were obtained by adding furimazine (10 μM) to a solution of labeled or unlabeled sensor protein constructs (1 nM) in HEPES (50 mM), NaCl (50 mM)(pH 7.2) buffer with or without MTX (100 μM). Spectra were measured with a step size of 2 nm and an integration time of 50 ms.

**Procedure for the templated unmasking of rhodamine by LUPIN**. The templated reactions were carried out in a 96 well black plates in 50 mM HEPES, 50 mM NaCl (pH 7.2) at 25 °C. The stock solutions of PNA probes (in deionized water or DMSO), LUPIN construct, and sodium ascorbate were diluted in the reaction buffer and then added to wells (100 μL/well). The reactions were initiated by the addition of 2 μL of an ethanolic solution of furimazine (5 mM) (final concentration 100 μM). Each experiment was performed in triplicates. The concentration of released rhodamine by LUPIN was calculated from the fluorescence (ex: 490 nm; em: 530 nm; cutoff: 515 nm) using a standard curve prepared with free rhodamine, ascorbate (10 mM), and furimazine (100 μM).

**Procedure for the templated unmasking of PNA-drug conjugates**. The template reactions to release drugs were performed as follows: solutions of PNA-Ru (10 μM) and PNA-PyDrug (Drug: Duocarmycin-OMe, Ibrutinib, or Raloxifene) (100 μM) in PBS (10 mM, pH 7.4) with sodium ascorbate (10 mM) were mixed in an Eppendorf. Samples were injected into LC-MS at 0, 1 or 2 min irradiation time points. Samples were irradiated with a collimated LED light (455 nm, 1 W: Thorlabs, part number M455L2-C1—www.thorlabs.com). Areas under the curve for the chromatograms were calculated using Thermo Xcalibur Qual Browser. This integration was used to estimate a yield understanding that this method underestimates the actual yield since the product has a lower absorption coefficient than the starting material.

*Cell culture:* MCF-7 and SKBR3 cell lines were obtained from the American Type Culture Collection (ATCC) and expanded following their instructions. MCF-7 cells were maintained in Dulbecco's modified Eagle medium (DMEM, Gibco) supplemented with 10% FCS and 1% pen-strep antibiotic at 37 °C under 5% CO$_2$ in a humidified incubator. SKBR3 cells were grown in Real McCoy's medium containing 10% FCS and 1% pen-strep. All cell lines were regularly tested for mycoplasma contamination by staining with Hoechst 33342.

**Ibrutinib competition in SKBR3 cells**. $5 \times 10^3$ SKBR3 cells were seeded in glass bottom dishes and grown for 24 h in a humidified incubator at 37 °C, 5% CO$_2$. Cells were then washed twice with DPBS containing magnesium and calcium. To the control cells, PNA-Py-Ibrutinib (**7**, 10 μM) in HBSS (0.1% BSA; with magnesium and calcium), incubated for 2 h at room temperature, was added to the cells and incubated for 30 min at 37 °C and 5% CO$_2$. The same was repeated for the LUPIN release, where PNA-Py-Ibrutinib (7, 10 μM), SNAP-(Ru-PNA-Mtx)NLuc-cpDHFR, 10 nM), sodium ascorbate (10 mM), and furimazine (100 μM) in HBSS (0.1% BSA; with magnesium and calcium), incubated for 2 h at room temperature, was added to the cells and incubated for 30 min at 37 °C and 5% CO$_2$. After 30 min, Ibrutinib-Cy3 (**9**, 50 nM) was added and the cells were further incubated for 30 min at 37 °C and 5% CO$_2$ after which the cells were washed twice with DPBS (with magnesium and calcium) and once with DMEM(-)(no phenol red). The cells were then imaged using a Leica SP8 fluorescent microscope with filter settings for Cy3 in the Leica software. Images were analyzed by image J.

**Duocarmycin-OMe and PNA-Py$^2$Duo toxicity in MCF-7 cells**. MCF-7 cells were seeded into 96-well plates (10$^4$ cells/well) and allowed to adhere overnight. Media was replaced with Duo-OMe (**4**) and PNA-Py$^2$Duo (**12**) at different concentrations in Leibovitz's medium (Gibco) and cells were incubated at 37 °C under 0% CO$_2$ in humidified incubator for 3 h. The cells were then washed three times with DMEM; fresh media (DMEM, no phenol-red) was replaced and the cells were incubated for additional 72 h at 37 °C, 5% CO$_2$. 10 μL of Hoechst 33342 from a 5 μg/mL stock in PBS was added to each well and incubated for 15 min at 37 °C. Fluorescence images were acquired using a HTS IXM microscope and subsequent image analysis and nuclei count was achieved by using MetaXpress® software. Drug effect was expressed as normalized nuclei count. 50% nuclei count was obtained from sigmoidal curve fits of normalized nuclei count vs. concentration data using Graph-Pad Prism 7. All experiments were conducted in triplicates, with error bars representing the standard error of the mean.

**Lupin release of Duocarmycin-OMe in MCF-7 cells**. MCF-7 cells ($10^4$ cells/well) were seeded in 96 well plates and left for 24 h. Cell media was aspirated and replaced with Leibovitz's medium, containing various concentrations of PNA-Py$^2$Duo (**12**) (0–250 μM) and furimazine (0–100 μM). Sodium Ascorbate was used at a concentration of 1 mM and SNAP-(Ru-PNA-Mtx)NLuc-cpDHFR was used at a concentration of 10 nM. The plates were then incubated at 37 °C for 3 h with no $CO_2$ after which the cells were gently washed repeatedly with DMEM, containing 10% FCS, 1% pen-strep and no phenol red and placed in the incubator (37 °C, 5% $CO_2$). After 72 h Hoechst 33342 stain was added at a concentration of 5 μg/μL followed by incubation for 30 min at 37 °C and 5% $CO_2$. Bright field and fluorescence images were then acquired with the IXM system with DAPI settings at 37 °C and 5% $CO_2$. Excitation = 100 ms. Objective 20×. Fluorescence images were analyzed using MetaXpress® software by using a nuclei count protocol were the particle mask size was width = 10 μm; height = 25 μm; fluorescence cutoff threshold = 1000. Each well was analyzed by the acquisition of matrix of images covering the well (3 × 3 tiles). The central tile is shown in Fig. 5.

### Data availability

The authors declare that the main data supporting the findings of this study are available within the article and its Supplementary Information files. Raw data has been deposited and is available (https://doi.org/10.5281/zenodo.1312221).

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

### Acknowledgements

This work was supported by the Swiss National Science Foundation and NCCR Chemical Biology. We thank Prof. Kai Johnsson for the kind gift of the LUCID plasmid construct (pET51b(+)SNAP-Pro30-NLuc-cpDHFR). We thank Dimitri Moreau for his assistance with the cytotoxicity assay.

### Author contributions

E.L., S.A. and N.W. conceived and designed experiments and analyzed the data. M.A. synthesized and tested the linker used for compound 12. E.L. and N.W. prepared the figures and wrote the paper.

### Additional information

**Competing interests:** The authors declare no competing interests.

