## [Peer Review File · Nature Communications]

Reviewers' comments:

Reviewer #1 (Remarks to the Author):

Lindberg and colleagues presents an interesting example of using emission from bioluminescence to trigger photo-uncaging for controlling small molecule activity. The concept of using bioluminescence to control photocatalysis is novel and have potentially wide use. However, as described below, it is unclear what the major enabling feature of the developed construct is. Moreover, innovativeness of the work is reduced since the SNAP-NLuc-cpDHFR construct has been previously developed. Also, the demonstration to control cytotoxicity of duocarmycin is inconclusive. For these reasons, the manuscript is not recommended for publication in Nature Communications. Detailed comments are below:

1. Using bioluminescence may enable new applications, since the luminescent molecule can be genetically encoded. Also, luminescence does not require optical excitation, which makes it an attractive means of detection under resource limited settings. However, for the demonstration that the authors showed in Figures 4 and 5, do not leverage these aspects. The experiments demonstrate the feasibility of the concept, but such they do not show what is new and unique about the construct developed.
2. As it is acknowledged, the SNAP-NLuc-cpDHFR has been developed by Griss and colleagues. Then, the novelty of this work is reduced to finding that a ruthenium-based photocatalyst can be activated by bioluminescence.
3. In the experiment to measure the cytotoxicity of duocarmycin (Fig. 5), it would be more conclusive to measure the cell viability, rather than counting the number of cells remaining. The drug may be lowering the adhesion of cells, not actually killing the cells.

Reviewer #2 (Remarks to the Author):

The paper describes an important extension of the application of bioluminescence resonance energy transfer (BRET) to trigger bioorthogonal reactions e.g. liberation of a drug in cell cultures interesting for a wide readership. This approach is new, robust and widely applicable. The data shown sufficiently demonstrate the applicability and usefulness of the system derived from Kai Johnsson LUCID-system. The system presented is highly modular and can be expected to be used for range of phototriggered bioorthogonal reactions. The insights given to the stability of the system and the improvements e.g. to strongly reduce „dark-release“ artifacts is convincing. Some of the following questions should be addressed:

Can the turnover-rate been modified by the intensity of light, or light pulses (has this been done already ?) to control the number of uncaged effector molecules e.g. for signaling purposes in chemical biology?

On page 6 in the summary the authors state: „The reaction based on an ruthenium....and shown to operate in a live organism.“ The last part „operate in a live organism“ is misleading since it implies that the whole system is uptaken by the cells whereas the system seem to be dissolved in the extracellular medium and liberates the target molecule extracellularly!

The same has to be stated the term „Lupin release of Duocarmycin-OMe in MCF-7 cells.“ on page S6 also misleading since the whole bioorthogonal reaction until liberation of the drug takes place outside the cell! No hints are given the authors as well as no experimental prove is presented to assume the uptake of the complete reaction system by cells! This would be highly relevant since timely liberation of a caged prodrug already exist extensively. It would be good, if the authors can

explicitly refer to the advantages and differences to show for which applications which systems is fruitful since the authors will address a rather wide readership.

If intracellular bioorthogonal reactions have been conducted the uptake of some components - the lupin-construct, MTX, PNA-drug conjugates... regarding time rate, general concentration dependence are missing.

If the questions above are addressed the paper should be ready for publication.

Reviewer #4 (Remarks to the Author):

The contribution by Lindberg and coworkers reports on the development of a new molecular system in which light produced by the luciferase NanoLuc is transferred to a ruthenium photocatalyst, which is then capable of uncaging pyridinium-PNA from a variety of small molecules. The ruthenium-PNA photocatalyst and its use in unmasking small molecule drugs using a combination of photocatalysis and PNA-hybridization-based substrate binding was previously reported by this group. Here this system was combined by the LUCID BRET-sensors developed previously in the group of Kai Johnsson. The authors convincingly showed that the principle works and that bioluminescence can be harvested to drive the ruthenium-controlled reactions. Moreover, because the distance between the luciferase and the ruthenium complex can be modulated by the LUCID-target molecule, in this case methotrexate, the efficiency of the uncaging reaction is also controlled by this presence of this small molecule. Overall the work is solid and the experiments are technically sound, involving a large body of synthesis and different experiments. My only major question is why one would want to develop a photocatalyst which is controlled by luciferase activity. The major efforts reported here on a number of cytotoxic drugs hint at a biological/medical application, but it is not clear what applications the authors eventually would have in mind. If it was to develop a small-molecule controlled photocatalyst for uncaging pro-drugs it would make more sense to have FRET-based control. As far as I can see it, this is a system which can be triggered by the presence of the luciferase substrate fumirazine, but it is not clear why and when that would be useful.

Other comments and questions:

1. the authors may also want to discuss and refer to some of the work reported to use BRET to induce photoswitching in the photoswitchable protein DRONPA (Zhang et al (2013) J Phys Chem Lett, 4, 3897-3902)
2. "... we calculated the Forster distance (R_0) to be 16 Å with an energy transfer efficiency (EFRET) of 0.64". What does this mean? I understand how one could calculate a Forster distance, but the energy transfer efficiency will then be dependent on the distance and orientation. By definition the energy transfer efficiency at the Forster distance is 50%.

Please find the revised manuscript (Luciferase-Induced photoreductive uncaging of small-molecule effectors: NCOMMS-18-05097-T) where we've addressed all the referee's comments and performed small edits to comply with the editorial check list. Please find a point-by-point response below.

Reviewer #1 (Remarks to the Author):

Lindberg and colleagues presents an interesting example of using emission from bioluminescence to trigger photo-uncaging for controlling small molecule activity. The concept of using bioluminescence to control photocatalysis is novel and have potentially wide use.

> We thank the referee for this positive comment.

However, as described below, it is unclear what the major enabling feature of the developed construct is. Moreover, innovativeness of the work is reduced since the SNAP-NLuc-cpDHFR construct has been previously developed. Also, the demonstration to control cytotoxicity of duocarmycin is inconclusive. For these reasons, the manuscript is not recommended for publication in Nature Communications. Detailed comments are below:

> As acknowledged above, this is the first demonstration of a BRET promoted chemical reaction. The fact that the SNAP-NLuc-DHFR has been previously reported does not take away from this novelty.

1. Using bioluminescence may enable new applications, since the luminescent molecule can be genetically encoded. Also, luminescence does not require optical excitation, which makes it an attractive means of detection under resource limited settings. However, for the demonstration that the authors showed in Figures 4 and 5, do not leverage these aspects. The experiments demonstrate the feasibility of the concept, but such they do not show what is new and unique about the construct developed.

> The text has now been reworked to discuss these points. We have not leverage the possibility of genetically encoding the BRET donor at this stage because the study focused on the novelty of the chemistry and the possibility to uncage effector molecules, for which we show three different examples. There are no precedents for a BRET-promoted chemical reaction and its scope and performance had to be investigated prior to any subsequent applications. The fact that LUPIN could ultimately be used in a genetically encoded system goes without saying but extends beyond the scope of this study.

2. As it is acknowledged, the SNAP-NLuc-cpDHFR has been developed by Griss and colleagues. Then, the novelty of this work is reduced to finding that a ruthenium-based photocatalyst can be activated by bioluminescence.

> The novelty of the work lies in the experimental demonstration that BRET can lead to a bioorthogonal chemical transformation and effector uncaging. We demonstrated that this required a specially designed synthetic linker in order to achieve a high enough concentration of substrate adjacent to the photocatalyst (templated chemistry). In its absence, the yield of the BRET-promoted reaction was dismal (Figure 2b, orange trace). The fact that this novel synthetic linker works on an existing BRET sensor without further engineering of the sensor suggest that it is broadly applicable and could be implemented in other reported BRET sensors without cumbersome engineering. We believe this is a positive point.

3. In the experiment to measure the cytotoxicity of duocarmycin (Fig. 5), it would be more conclusive to measure the cell viability, rather than counting the number of cells remaining. The drug may be lowering the adhesion of cells, not actually killing the cells.

> The screens shown in Fig. 5 used a high-content image-based method to quantitatively assess cytotoxicity and differentiate between cytotoxicity and growth inhibition. The advent of automated microscopes for HTS has made this protocol the methodology of choice and benefits from increased dynamic range and accuracy compared to traditional viability screens with MTT or WST-1 where one essentially measures mitochondrial

activity. Gravey et al Scientific Reports, 6:29752 (2016) has an extended discussion on the virtue of high-content image-based method for viability screens. Traditional cell viability assays require washing step and would not discriminate between a drug that promotes cellular detachment vs a cytotoxic drug. The presence of high level of cellular debris in the high-content image-based method does and is indeed what is observed in the experiments and shown in Fig 5. However, this is somewhat beside the point because the mode of action of duocarmycin is well characterized and the compound is known to be a highly cytotoxic drug. The IC50 that we measure with the free drug are consistent with literature reports. The control experiments shown in Fig 5 lacking either the prodrug or furimazine are comparable to the untreated cells.

Reviewer #2 (Remarks to the Author):

The paper describes an important extension of the application of bioluminescence resonance energy transfer (BRET) to trigger bioorthogonal reactions e.g. liberation of a drug in cell cultures interesting for a wide readership. This approach is new, robust and widely applicable. The data shown sufficiently demonstrate the applicability and usefulness of the system derived from Kai Johnsson LUCID-system. The system presented is highly modular and can be expected to be used for range of phototriggered bioorthogonal reactions. The insights given to the stability of the system and the improvements e.g. to strongly reduce „dark-release“ artifacts is convincing.

> We thank the referee for these positive comments.

Some of the following questions should be addressed:

Can the turnover-rate be modified by the intensity of light, or light pulses (has this been done already ?) to control the number of uncaged effector molecules e.g. for signaling purposes in chemical biology?

> This is an excellent point. Indeed, there is a direct correlation between photon flux (furimazine added) and yield of product. The text has been amended to reflect this point: “Further evaluation of our LUPIN system revealed there is a direct correlation between the photo flux emitted from Nluc and the amount of uncaged product (Supplementary Fig. 6a–d reactions at different furimazine concentrations)”. The fact that at lower furimazine concentration, Nluc produce a burst of luminescence suggest that pulsing product release should be possible.

On page 6 in the summary the authors state: „The reaction based on an ruthenium....and shown to operate in a live organism.“ The last part „operate in a live organism“ is misleading since it implies that the whole system is uptaken by the cells whereas the system seem to be dissolved in the extracellular medium and liberates the target molecule extracellularly!

> We thank the referee for pointing out this confusing statement. The statement was in reference to a prior publication where ruthenium-catalyzed photoredox chemistry has been performed live organism (ref 8). The sentence has been removed from the conclusion.

The same has to be stated the term „Lupin release of Duocarmycin-OMe in MCF-7 cells.“ on page S6 also misleading since the whole bioorthogonal reaction until liberation of the drug takes place outside the cell! No hints are given the authors as well as no experimental prove is presented to assume the uptake of the complete reaction system by cells! This would be highly relevant since timely liberation of a caged prodrug already exist extensively. It would be good, if the authors can explicitly refer to the advantages and differences to show for which applications which systems is fruitful since the authors will address a rather wide readership.

> We appreciate this constructive criticism and agree that the wording is misleading. The wording has been changed to “LUPIN release of duocarmycin in cell culture”. The chemistry was performed extracellularly and we have not studied the cellular uptake at this stage.

If the questions above are addressed the paper should be ready for publication.

Reviewer #4 (Remarks to the Author):

The contribution by Lindberg and coworkers reports on the development of a new molecular system in which light produced by the luciferase NanoLuc is transferred to a ruthenium photocatalyst, which is then capable of uncaging pyridinium-PNA from a variety of small molecules. The ruthenium-PNA photocatalyst and its use in unmasking small molecule drugs using a combination of photocatalysis and PNA-hybridization-based substrate binding was previously reported by this group. Here this system was combined by the LUCID BRET-sensors developed previously in the group of Kai Johnsson. The authors convincingly showed that the principle works and that bioluminescence can be harvested to drive the ruthenium-controlled reactions. Moreover, because the distance between the luciferase and the ruthenium complex can be modulated by the LUCID-target molecule, in this case methotrexate, the efficiency of the uncaging reaction is also controlled by this presence of this small

molecule. Overall the work is solid and the experiments are technically sound, involving a large body of synthesis and different experiments.

> We thank the referee for these positive comments.

My only major question is why one would want to develop a photocatalyst which is controlled by luciferase activity. The major efforts reported here on a number of cytotoxic drugs hint at a biological/medical application, but it is not clear what applications the authors eventually would have in mind. If it was to develop a small-molecule controlled photocatalyst for uncaging pro-drugs it would make more sense to have FRET-based control. As far as I can see it, this is a system which can be triggered by the presence of the luciferase substrate fumirazine, but it is not clear why and when that would be useful.

> The catalyst has a fairly broad absorption for its excitation (from 350 to 470 nm) which would require a fluorophore well into the short wavelength UV to achieve a FRET-based system. UV light has very poor penetration and is toxic. The methodology reported does not require to bring light and hence the reaction can be performed in condition where direct excitation would be problematic. Since the sensor (or a simpler Nluc-SNAP construct) can be genetically encoded, the reaction could also be constrained to specific cell types. While this extends beyond the present manuscript, we are currently investigating the use of this technology to achieve effector uncaging with subcellular resolution. This would be exceptionally difficult to achieve with direct irradiation. We demonstrate the conditional unmasking of drugs and we expect that the technology could become useful in conditional and targeted therapy (smart therapeutics). The ability to engineer logic gates in synthetic biological network is also important and the methodology offers a new means to do so with an abiotic reaction. The conclusion has been reworked to clarify this message.

Other comments and questions:

1. the authors may also want to discuss and refer to some of the work reported to use BRET to induce photoswitching in the photoswitchable protein DRONPA (Zhang et al (2013) J Phys Chem Lett, 4, 3897-3902)

> The reference has now been included.

2. “.. we calculated the Forster distance (R_0) to be 16 Å with an energy transfer efficiency (EFRET) of 0.64”. What does this mean? I understand how one could calculate a Forster distance, but the energy transfer efficiency will then be dependent on the distance and orientation. By definition the energy transfer efficiency at the Forster distance is 50%.

> The Forster distance is defined as the distance corresponding to a 50% energy transfer. The calculated energy transfer efficiency is 0.64, meaning that the distance between the emitting fumirazine and ruthenium photocatalyst is shorter than 16Å. The details of the equations used to calculate these figure are on page 2 of the supplementary information. The text has been amended clarity: “we calculated the Förster distance (R_0 , distance corresponding to 50% energy transfer) to be 16 Å with an energy transfer efficiency (ERET) of 0.64 (see supplementary information details)”.

Reviewers' comments:

Reviewer #1 (Remarks to the Author):

As the authors have said, the study is first to show control of photocatalysis using BRET. In my opinion, this new capability will be useful only if it can be genetically encoded and functional inside live cells. Otherwise, there are other similar ways to control photocatalysis that may be easier to use and more efficient. However, it should be acknowledged that the convincing demonstration of controlling BRET induced photocatalysis is a significant progress. Therefore, I recommend publishing the article in Nat. Comm.

Reviewer #2 (Remarks to the Author):

The decision for the paper is certainly on an edge. On the one hand site the first demonstration of a BRET-promoted chemical reaction is interesting due to the complex molecular assemble allowing to influence the system at each level of these various system components and parameters. Certainly a future application in vivo would lever the system for some interesting systematic uses to triggered photocatalytic events and potential señor function sin response to a small chemical molecules. Thus, the genetic encoding of the luminescent molecules opens interesting ways for chemical, synthetic and systems biology.

The current focus of the paper is concerned with the demonstration of the chemical possibilities. They, from a system-complexity point of view, are remarkable, but do not easily allow to be of major advantage over other in vitro systems in term s of their potential to chemically or optically control a small molecule liberation system.

In my opinion a positive judgement for the publication of the paper in its current scope in NatCom would require a positive judgement for its effective in vivo application in the future. Since the current system contains some components (PNA...) and system assembly steps challenging to be facilitated completely in vivo based on a completely genetically encoded system, one has to assume a main application of the system to be restricted to in vitro applications currently. Thus, I do not recommend the paper for publication in Nat.Com.

Reviewer #4 (Remarks to the Author):

I am satisfied with the response to most of my questions and those of the other reviewers. I believe the manuscript can be accepted for publication, but would like the authors to address the following minor remaining issues:

- protein (Dronap) to a ON state -> protein (Dronpa) to an ON state

- "we calculated the Förster distance (R_0 , distance corresponding to 50% energy transfer) to be 16 Å with an energy transfer efficiency (ERET) of 0.64 (see supplementary information details)". I still think this is not very clear. Why not 'Based on the quantum yield of Nanoluc and the spectral overlap of the Nanoluc emission spectrum with the excitation spectrum of the Ru-complex, we calculated the Forster distance to be 16 Å. Since the energy transfer efficiency between NanoLuc and th Ru-complex was found to be 0.64, this means that the average distance needs to be less than 16 Å.' The authors may want to note that this is actually quite a surprising result, since such as small distance suggests that the Ru-complex is very close to the Nanoluc surface, possibly even forming a transient complex with it. Is this conceivable?

- 'BRETs have frequently been utilized as reporting modalities in diverse sensors.' What are BRETs? Shouldn't this be 'BRET has ..'

RE: Luciferase-Induced photoreductive uncaging of small-molecule effectors: NCOMMS-18-05097-T

Based on the referee responses to the revised manuscript (NCOMMS-18-05097A), I gather the rejection decision is based on reviewer #2's comments. Both reviewer #1 and reviewer #4 recommend publication (reviewer #1 : "it should be acknowledged that the convincing demonstration of controlling BRET induced photocatalysis is a significant progress. Therefore, I recommend publishing the article in Nat. Comm"; reviewer #4: "I am satisfied with the response to most of my questions and those of the other reviewers. I believe the manuscript can be accepted for publication, but would like the authors to address the following minor remaining issues" (see below for full comments). Reviewer #2's (full comments pasted below) thinks the decision is on an edge. Reviewer #2 start with positive comments regarding the novelty of BRET-promoted chemical reaction and interesting applications in chemical, synthetic and system biology, however fails to see utility if the system is not genetically encoded and states "In my opinion a positive judgement for the publication of the paper in its current scope in NatCom would require a positive judgement for its effective in vivo application in the future. Since the current system contains some components (PNA...) and system assembly steps challenging to be facilitated completely in vivo based on a completely genetically encoded system, one has to assume a main application of the system to be restricted to in vitro applications currently. Thus, I do not recommend the paper for publication in Nat.Com." These reservations were not part of reviewer #2's comment in the 1st round of reviewing and are frankly unfounded. In the first round, reviewer #2 stated that "The paper describes an important extension of the application of bioluminescence resonance energy transfer (BRET) to trigger bioorthogonal reactions e.g. liberation of a drug in cell cultures interesting for a wide readership. This approach is new, robust and widely applicable. The data shown sufficiently demonstrate the applicability and usefulness... If the questions above are addressed the paper should be ready for publication " (see full comments below), with minor point that were all addressed in the revisions.

I am thus appealing the rejection decision based on the fact the argument of reviewer #2 in the second round are unfounded. PNA-templated chemistry using a photocatalytic ruthenium reaction with external light has already been shown to operate in a live organism (zebrafish: Nucleic Acid Templated Chemical Reaction in a Live Vertebrate ACS. Cent. Sci. 2016, 2, 394, reference 16 in the manuscript), hence the "system assembly steps" have been demonstrated in vivo. Quoting from the conclusion of ref 16: "In summary, we have described a nucleic acid templated chemical reaction [using PNA] that reports on specific RNA sequences in a live vertebrate thus providing a rapid and simple platform to image miRNA.". Furthermore, applications do not require that the bioluminescent protein be genetically encoded. One could certainly envision to conjugate or fuse the bioluminescent protein to a targeting unit in order to uncage a drug at a site of interest, akin to antibody drug conjugates but with higher payload by virtue of the catalysis. In the present manuscript, we further demonstrate that a logic gate can be included in the system, with a response conditional on an analyte. Last but not least, PNA-templated reactions can be engineered with a chirality that precludes DNA or RNA hybridization and bypass any complication from cross talk resulting from hybridization to endogenous nucleic acids (ref 17 in the manuscript). While PNA-templated chemistry had been demonstrated in vivo (ref 16), there remains a challenge to bring homogeneous irradiation at 450 nm wave length to deep tissue. Thus, the ability to achieve photocatalytic reaction with bioluminescence is an important advance, be it with an administered bioluminescent protein or engineered cells. This is indeed the conclusion of reviewer #1 and #4. I would like to add that there is mounting interest in photoswitchable drug (For example, Photoswitchable Inhibitors of Microtubule Dynamics Optically Control Mitosis and Cell Death, Trauner et al, Cell 2015, DOI: 10.1016/j.cell.2015.06.049) and the present manuscript is relevant to this burgeoning field that is also limited ultimately by the lack of homogenous and deep penetration of blue-green light.

Reviewer #2 (Remarks to the Author) - first revision:

The decision for the paper is certainly on an edge. On the one hand site the first demonstration of a BRET-promoted chemical reaction is interesting due to the complex molecular assemble allowing to influence the system at each level of these various system components and parameters. Certainly a future application in vivo would lever the system for some interesting systematic uses to triggered photocatalytic events and potential señor function sin response to a small chemical molecules. Thus, the genetic encoding of the luminescent molecules opens interesting ways for chemical, synthetic and systems biology.

The current focus of the paper is concerned with the demonstration of the chemical possibilities. They, from a system-complexity point of view, are remarkable, but do not easily allow to be of major advantage over other in vitro systems in terms of their potential to chemically or optically control a small molecule liberation system.

In my opinion a positive judgement for the publication of the paper in its current scope in NatCom would require a positive judgement for its effective in vivo application in the future. Since the current system contains some components (PNA...) and system assembly steps challenging to be facilitated completely in vivo based on a completely genetically encoded system, one has to assume a main application of the system to be restricted to in vitro applications currently. Thus, I do not recommend the paper for publication in Nat.Com.

Reviewer #2 (Remarks to the Author) – original submission:

The paper describes an important extension of the application of bioluminescence resonance energy transfer (BRET) to trigger bioorthogonal reactions e.g. liberation of a drug in cell cultures interesting for a wide readership. This approach is new, robust and widely applicable. The data shown sufficiently demonstrate the applicability and usefulness of the system derived from Kai Johnsson LUCID-system. The system presented is highly modular and can be expected to be used for range of phototriggered bioorthogonal reactions. The insights given to the stability of the system and the improvements e.g. to strongly reduce „dark-release“ artifacts is convincing. Some of the following questions should be addressed:

Can the turnover-rate be modified by the intensity of light, or light pulses (has this been done already ?) to control the number of uncaged effector molecules e.g. for signaling purposes in chemical biology?

On page 6 in the summary the authors state: „The reaction based on an ruthenium....and shown to operate in a live organism.“ The last part „operate in a live organism“ is misleading since it implies that the whole system is uptaken by the cells whereas the system seem to be dissolved in the extracellular medium and liberates the target molecule extracellularly!

The same has to be stated the term „Lupin release of Duocarmycin-OMe in MCF-7 cells.“ on page S6 also misleading since the whole bioorthogonal reaction until liberation of the drug takes place outside the cell! No hints are given the authors as well as no experimental prove is presented to assume the uptake of the complete reaction system by cells! This would be highly relevant since timely liberation of a caged prodrug already exist extensively. It would be good, if the authors can explicitly refer to the advantages and differences to show for which applications which systems is fruitful since the authors will address a rather wide readership.

If intracellular bioorthogonal reactions have been conducted the uptake of some components - the lupin-construct, MTX, PNA-drug conjugates... regarding time rate, general concentration dependance are missing.

If the questions above are addressed the paper should be ready for publication.

Reviewer #4 (Remarks to the Author) – first revision:

I am satisfied with the response to most of my questions and those of the other reviewers. I believe the manuscript can be accepted for publication, but would like the authors to address the following minor remaining issues:

- protein (Dronap) to a ON state -> protein (Dronpa) to an ON state

- “we calculated the Förster distance (R_0 , distance corresponding to 50% energy transfer) to be 16 Å with an energy transfer efficiency (ERET) of 0.64 (see supplementary information details)”.

I still think this is not very clear. Why not ‘Based on the quantum yield of NanoLuc and the spectral overlap of the NanoLuc emission spectrum with the excitation spectrum of the Ru-complex, we calculated the Forster distance to be 16 Å. Since the energy transfer efficiency between NanoLuc and th Ru-complex was found to be 0.64, this means that the average distance needs to be less than 16 Å.’ The authors may want to note that this is actually quite a surprising result, since such a small distance suggests that the Ru-complex is very close to the NanoLuc surface, possibly even forming a transient complex with it. Is this conceivable?

-‘BRETs have frequently been utilized as reporting modalities in diverse sensors.’ What are BRETs? Shouldn't this be ‘BRET has ..’

All the suggested changes have been made.